# Underlying Ciliary Body Uveal Melanoma in a Patient with Chronic Lymphocytic Leukemia Presenting for Hyphema

**DOI:** 10.3390/diagnostics12061312

**Published:** 2022-05-25

**Authors:** Mihai Adrian Păsărică, Paul Filip Curcă, Christiana Diana Maria Dragosloveanu, Cătălina Ioana Tătaru, Ioana Roxana Manole, Gabriela Elisabeta Murgoi, Alexandru Călin Grigorescu

**Affiliations:** 1Clinical Department of Ophthalmology, Carol Davila University of Medicine and Pharmacy, 020021 Bucharest, Romania; m.pasarica@yahoo.com (M.A.P.); christianacelea@gmail.com (C.D.M.D.); catalina_tataru@yahoo.com (C.I.T.); alexgrigorescu2004@yahoo.com (A.C.G.); 2Department of Ophthalmology, Clinical Hospital for Ophthalmological Emergencies, 010464 Bucharest, Romania; ioana.manole94@yahoo.com; 3Department of Oncology, Institute of Oncology Prof. Dr. Alexandru Trestioreanu, 022328 Bucharest, Romania; gabrielamurgoi@yahoo.com; 4Department of Oncology, Clinical Hospital of Nephrology Dr. Carol Davila, 010731 Bucharest, Romania

**Keywords:** ciliary body melanoma, uveal melanoma UM, chronic lymphocytic leukemia CLL, hyphema, immunosuppression

## Abstract

(1) Background: Ciliary body uveal melanoma is a rare subtype of uveal melanoma which comprises 3–5% of melanomas, an immunogenic cancer, and can present multifaceted initial clinical manifestations, masquerading as various ocular pathologies. Chronic lymphocytic leukemia (CLL) presents immunodeficiency and risk for the development of a secondary malignancy, with Bruton’s tyrosine kinase inhibitor treatment having a mutagenic effect and a secondary anti-platelet aggregation effect. We present the case of a 65-year-old patient undergoing treatment for CLL with ibrutinib who presented with recurrent hyphema that masked an underlying, inferiorly situated, ciliary body uveal melanoma; (2) Methods: Retrospective case review; (3) Results: An ophthalmological examination together with imaging via mode B ultrasound and contrast-enhanced magnetic resonance imaging resulted in the clinical and imagistic diagnosis of a ciliary body uveal melanoma. A pathological examination of the enucleated eye confirmed the diagnosis. Postoperative tumoral reoccurrence was not detected for 1½ years, however, CLL immunosuppression worsened with admission for severe COVID-19 disease. (4) Conclusions: CLL patient screening for melanoma should also include detailed ophthalmological examinations, which could also include ultrasound ophthalmological imaging. The avoidance of uveal melanoma metastatic disease is paramount for patient survival. CLL manifests additional profound immunosuppression.

## 1. Introduction

Chronic lymphocytic leukemia (CLL) is the most common form of leukemia in developed countries with an incidence of 4–5 per 100,000 individuals [1]. CLL is a chronic lymphoproliferative disorder characterized by the accumulation of monoclonal, mature, CD5+ cells in the peripheral blood, bone marrow and secondary lymphoid organs [2]. Diagnosis is established by an elevated B-cell count of at least 5000 cells per microliter of blood with characteristic immunophenotyping of cells co-expressing CD5 and B-cell surface antigens CD19, CD20 and CD23 [1] and specific low expression of surface immunoglobulin, CD19, CD20 and CD79b [1]. Each clone of leukemia cells expresses either κ or λ immunoglobulin chains. The clinical presentation of the disease is highly variable, from asymptomatic and indolent disease that may never require therapy to active complicated disease [3], with staging and risk stratification for CLL following the Rai and Binet systems [1]. Since healthy B-cells are critically essential for both B-cell immune response and the polarization of an effective T cell response [4], CLL patients associate profound immunosuppression [4] with malignant B-cells evading immune detection by inducing T cell anergy and improper T-helper2 (Th-2) polarization, leading to recurrent infections [3] and T cell immune dysfunction [4]. The treatment of CLL has been fundamentally changed by the introduction of kinase inhibitors targeting B-cell receptor signaling kinases such as first generation Bruton’s tyrosine kinase (BTK) inhibitor ibrutinib and the second generation more selective co-valent BTK inhibitor acalabrutinib [3]. Patients with CLL present a higher risk of bleeding [3], and those undergoing ibrutinib treatment present inhibition of BTK-dependent platelet aggregation [3], with a more common occurrence of low-grade bleeding (35% vs. 15%) [3] and major hemorrhage (4.4 vs. 2.8%) [3] versus comparative therapy. Furthermore, established treatment with BTK could present a mutagenic risk [5] with increased risk for the development of a second malignancy.

Melanoma is considered an immunogenic cancer [6,7] with immunocompromised populations at higher risk [7]. An increased incidence of melanoma in CLL patients has been reported in patients with CLL [6,7,8] with different meta-analyses showing a two-fold (2.07) [8] or four-fold increase in risk compared to the general population [7], strongly suggesting routine screening for melanoma in CLL patients [7,8]. Uveal melanoma (UM) accounts for only 3–5% of all melanomas [9,10] and represents the most common primary malignancy affecting the eye [9] which can arise anywhere in the uveal tract (iris, ciliary body and choroid), often involving multiple uveal structures [9]. Ciliary body melanoma represents an even rarer subtype, encompassing only 5 to 8% of UM cases. In comparison to cutaneous or mucosal melanomas, UM presents a different molecular profile, with the common chromosomal abnormalities encountered being 3-monosomy, 1-loss, 1q gain, 6q loss, 6p gain, 8p loss and 8q gain [10]. Due to localization inside the eye, lymphatic spread is exceedingly rare [9] and UM is approachable with globe-preserving therapies (radiation therapy, iodine-125 brachytherapy) [11] or curative surgical enucleation treatment, however, once metastatic disease develops by hematogenous dissemination, commonly involving the liver (93%), lung (24%) or bones (16%) [9], prognosis is unfavorable with median survival rates ranging from 4 to 15 months [11]. While various treatment approaches similar to cutaneous melanoma have been evaluated, such as systemic chemotherapy, immunotherapy with immune checkpoint inhibitors targeting the T-lymphocyte-associated antigen 4 (CTLA-4) (ipilimumab, tremelimumab) [11] or programmed cell death-1 (PD-1) (nivolumab, pembrolizumab), and targeting of the soluble T cell receptor (TCR) with IMCgp100, unfortunately UM does not respond as well as cutaneous melanoma to treatment and no consensus has yet been reached for its care, with prevention of metastatic disease remaining the most effective option [11].

## 2. Materials and Methods

Retrospective case review examining the patient’s presentation and previous medical history.

## 3. Results

### Case Presentation

A 65-year-old male Caucasian patient, currently undergoing treatment for chronic lymphocytic leukemia Rai-II and chronic hepatitis B, presented at the emergency department of our hospital with rapidly progressive loss of visual acuity in the right eye.

A complete ophthalmological and slit-lamp exam of the right eye revealed corneal oedema, a medium depth anterior chamber with hyphema occupying the inferior half, normal aspect of the visible iris stroma, reflexive pupil, nuclear and subcapsular cataract. The left eye presented a nuclear cataract and was otherwise unremarkable. The patients uncorrected visual acuity at presentation was 20/400 (0.05 decimal/1.3 logMAR) in the right eye and 20/40 (0.5 decimal/0.3 logMAR) in the left eye. Intraocular pressure (IOP) in both eyes measured 14 mmHg. Posterior pole examination was obstructed by the hyphema in the right eye and was normal in the left eye.

Upon further questioning, the patient confirmed that he was currently undergoing treatment for chronic lymphocytic leukemia with ibrutinib (Imbruvica), chronic hepatitis B with entecavir and tenofovir, type II diabetes mellitus and arterial hypertension. The patient recounted a previous presentation for bleeding in his right eye, and a search of hospital records confirmed the patient underwent treatment for hyphema 3 months prior to the current admission and was lost to follow-up. A presumptive diagnosis of recurrent hyphema possibly related to the patient’s chronic lymphocytic leukemia was established and the patient was admitted to our department for treatment and further etiological investigations. After obtaining samples for bloodwork, systemic treatment was commenced with intravenous (iv) I mannitol 200 mg/mL, II etamsylate 250 mg and II vitamin C 750 mg per day, and local treatment with topical non-steroidal anti-inflammatory drug (NSAID) diclofenac 4/day and mydriatics phenylephrine and tropicamide both 4/day. Subsequent bloodwork revealed hyperglycemia of 154 mg/dL, prolonged thromboplastin time of 15.6 s, thrombocytopenia with 133.000 platelets per microliter, increased mean platelet volume (MPV) of 11.2 and mean corpuscular volume of 98.1 and increased red cell distribution width (RDW) of 16.7.

After 2 days of medical treatment, a partial remission of the hyphema was achieved with subjective improvement in the patient’s vision. However, a careful slit-lamp biomicroscopy examination of the iris stroma, previously obstructed by the hyphema, revealed an irregular dark-pigmented mass involving the iris stroma and root, with adjacent corneal oedema, arousing suspicion of uveal melanoma. Furthermore, an enlarged conjunctival blood vessel was visible inferiorly and could be a sentinel episcleral vessel (Figure 1).

The slit-lamp biomicroscopy examination was repeated after the hyphema receded, at the 1-week follow-up (Figure 2). The normal iris stroma was absent inferonasal, having instead been replaced by the pigmented mass at this level. A likely sentinel episcleral vessel was visible. A gonioscopy examination, considered the clinical reference standard in assessing the anterior chamber angle [12], was performed using a three-mirror gonioscopy lens (Figure 3) (Ocular Instruments, Bellevue, Washington State, United States of America). The tumoral mass was present at the inferonasal iris and anterior chamber angle level. The superotemporal angle presented a normal, Shaffer grade 3, open angle aspect [13].

Mode B ultrasound scan (ABSolu Ultrasound, Quantel Medical, Cournon d’Auvergne, France), performed during the patient’s admission, confirmed the presence of an intraocular tumor, located in the inferior nasal sector, exhibiting medium reflectivity with an anterior-posterior diameter of 11.54 mm and surface measurement 95.97 mm^2^, highly suggestive for ciliary body uveal melanoma (Figure 4). Using mode A ultrasound scan (ABSolu Ultrasound, Quantel Medical, Cournon d’Auvergne, France), the tumor diameter was measured at 11.84 mm (C3) and the angle of ultrasonic absorption was 62.5° (Figure 5).

The patient was urgently referred to the ophthalmic oncology department. Contrast-enhanced magnetic resonance imaging (CE-MRI) (Figure 6) was performed with the substances administered in the following sequences: T1 sagittal, T2 axial, coronal and sagittal fluid attenuated inversion recovery (FLAIR), T2 three-dimensional constructive interference in steady state (3D-CISS), diffusion-weighted imaging (DWI), T2 axial gradient echo (GRE), T2 coronal with thin slices, T2 turbo inversion recovery magnitude (TIRM) dark fluid, T2 turbo spin echo (TSE) fat-suppressed (FS) coronal with thin slices, T1 axial volumetric interpolated breath-hold examination (VIBE) FS with iv. contrast, T1 coronal FS with iv. contrast and T1 multiplanar reformation/reconstruction (MPR). The intraocular mass presented intense and homogenous contrast capture, with maximum diameters of 17/12 mm axial and 10 mm vertical. No invasion of adjacent orbital tissues could be found. A frontal osteoma measuring 11/12/6.5 mm was also evidenced by CE-MRI, along with supratentorial cortical atrophy, with 2–3 hypersignaled microspots appearing in T2 and FLAIR, having a maximum diameter of 3.3 mm, without restriction in water diffusion or contrast capture and maxillary retention cysts measuring 26 mm on the right and 30 mm on the left. Based on the clinical presentation of the patient and imagistic investigations, a diagnosis of ciliary body melanoma was established.

Staging was reported as T2bNxM0 [14] according to computed tomography (CT) total body scan. The following enlarged lymph nodes were identified: right cardio-phrenic adenopathy, superior phrenic adenopathy with short axis under 8 mm, lombo-aortic and several mediastinal (<6 mm) enlarged lymph nodes. In our case, the patient’s chronic lymphocytic leukemia (CLL) comorbidity made assessing the presence of lymph node metastasis especially difficult due to enlarged lymph nodes caused by CLL. CT total body did not evidence hepatic metastasis.

In our case, the oncological surgeon identified the following risk factors for our patient: extensive local iris invasion which had repeatedly produced hyphema in a time span of 3 months (the first hyphema was in September, and the following episode was in December of the same year), an increased risk of hematogenous dissemination via the well-vascularized iris, tumor size (according to ultrasound measurement: the scan measured 11.84 mm), patient age (65 years old), difficulty in evaluating lymph node metastasis due to uveal melanoma in the presence of the patient’s CLL and inherent immunosuppression due to CLL and treatment with ibrutinib which could not be discontinued [15,16]. Considering the abovementioned risk factors, the oncological surgeon recommended enucleation as the treatment option, and after presenting all the therapeutic options and associated risks to the patient, enucleation of the right globe was preferred and performed.

A pathological examination of the excised specimen was performed. The enucleated right eye had a diameter of 25 mm. After sectioning, the macroscopic aspect was of a pigmented tumor extending supero-anterior with invasion of the uveal tract. The largest tumor diameter was also the basal diameter of 12 mm, and the apical height was 10 mm. The distance between the tumor and the optical nerve was 12 mm, without invasion of the optic nerve.

Histopathological microscopic analysis of the specimen confirmed infiltration by epithelioid tumoral cells and melanophages (Figure 7) in accordance with the diagnosis of ciliary body melanoma.

A predominant malign melanocytic proliferation of intermediate morphology cells (in between spindle type B and epithelioid, approximately 50%) was present, with a secondary proliferation of type B spindle cells (40%) and epithelioid cells (10%) (Figure 8). In accordance with the American Joint Committee of Cancer (AJCC) classification [14,17], our case presented a mixed cell type. The tumor cells were arranged into nests which were separated by fibrous conjunctival septa. Melanic pigmentation was present and relatively uniform intratumorally. A minimal chronic lymphocytic infiltrate was present, with melanophages situated mostly peritumorally. The proliferation of tumor cells had invaded the ciliary body, pars plana and choroid, with extension into the posterior chamber and secondary retinal detachment (Figure 8). Lympho-vascular or neural invasion was not detected. Maximal basal diameter was measured at 12 mm, maximum height at 9 mm. Staging was evaluated at pT2bNx [14].

Immunostaining was performed using the melanoma cocktail human melanoma black 45 (HMB45) and melanoma-associated antigen recognized by T cells (MART-1) clones M2-7C10 + M2-9E3 for melanoma screening, protein S-100 (EP32) antibody, anti-SRY-box transcription factor 10 (SOX10) antibody (SP267), Ki67 antigen (p3-NS1-Ag4-1) (Figure 9). According to Russo et al. [18] S100, SOX10 and HMB45 are significant markers of melanocytic differentiation and were positive in our case.

The patient has since attended regular follow-up screening for the past one and a half years without tumoral recurrence, however was then lost to follow-up. Further efforts in contacting the patient’s relatives revealed the patient was admitted for severe coronavirus (COVID-19) disease and as such was unable to further attend follow-up. Upon further inquiry the patient benefited from preventive messenger ribonucleic acid (mRNA) COVID-19 vaccination using a three-dose regimen, however, was unable to develop a sufficiently protective immune response in the immunosuppressive context of the patient’s CLL.

## 4. Discussion

Uveal melanoma is an ocular malignancy that can present a multifaceted clinical manifestation based on localization and patient comorbidities, “masquerading” as spontaneous intraocular bleeding in the anterior chamber (hyphema) [19,20,21,22] or involving the posterior pole as traumatic choroidal hematoma [23] or hemorrhagic choroidal detachment [24], or taking on the clinical presentation of various ocular pathologies such as chronic intraocular inflammation [25], pigment dispersion glaucoma [26] or secondary glaucoma [27], scleritis [28] or even orbital cellulitis [29]. In our case, the presumptive diagnosis was recurrent hyphema possibly related to coagulation abnormalities inherent in the patient’s chronic lymphocytic leukemia, with the patient’s treatment with BTK inhibitor ibrutinib also inhibiting BTK-dependent platelet aggregation. Subsequent lab work revealed mildly prolonged thromboplastin time and mild thrombocytopenia. Due to the inferior location of the tumor and gravitational pooling of blood in the anterior chamber, a clear visualization of the mass via slit-lamp examination was not initially possible. Considering the presence of CLL comorbidity, we admitted the patient for thorough medical supervision and imagistic investigations, suspecting a CLL involvement of the eye or a possible intraocular malignancy. After the hyphema partly receded, the following clinical aspects maximized the suspicion of ciliary body melanoma: an irregular dark-pigmented mass involving the iris stroma and root and adjacency signs such as corneal oedema and likely sentinel vessel [30]. The next step was ascertaining imagistic confirmation, with mode B ultrasound confirming the characteristics of a large ciliary body melanoma. Ossoinig [31] described four cardinal acoustic hallmarks of malignant melanoma on A-scan including a regular internal structure with similar height of the inner tumor spikes or regular decrease in height (positive angle kappa sign), low to medium reflectivity, solid consistency with no after movement of tumor spikes, and echographic sign of vascularization with a fast, spontaneous, continuous flickering vertical motion of single tumor spikes [32] and according to Minning CA et al. [33], an ultrasonic angle of absorption between 40° and 65° degrees is a criteria for the ultrasound diagnosis of malignant melanomas, with the angle 62.5° degrees, in our case. Another option for visualizing the anterior chamber and iris is ultrasound biomicroscopy (UBM) [34,35,36] which offers superior resolution and is helpful in assessing smaller-sized anterior segment tumors [35]. After obtaining imagistic confirmation, the patient was urgently referred to specialized ophthalmic oncology care, where CE-MRI and CT total body scans were performed for confirmation, staging, and ascertaining the presence of orbital extension or metastatic disease. Noteworthy to our case, the patient’s CLL can also produce enlarged lymph node aspects on imagistic investigations. The final clinical and imagistic diagnosis was established by the joint effort of ophthalmology and oncology specialists.

An assessment of available therapeutic options was performed.

The first assessed options were cyber-knife or gamma-knife stereotactic radiosurgery, teletherapy that provides high therapeutic doses of radiation while sparring adjacent tissues [32], however, the large size and extension of the tumor taken into consideration with the patient’s risk factors would have implied an unacceptably high risk of insufficient disease control and later metastatic spread. Furthermore, Langmann G et al. [37] noted that neovascular glaucoma can develop in patients after undergoing gamma knife radiosurgery for uveal melanomas located near the ciliary body and advised that such tumors be avoided and that the radiotherapy prescription dose be reduced to 40 Gy, a lower possible dose compared to that used for tumors located more posteriorly [37].

According to the COMS randomized trial of iodine 125 brachytherapy for choroidal melanoma [38], the mortality rates among patients treated with eye-conserving I-125 brachytherapy were similar to rates among enucleated patients for up to 12 years of follow-up and the COMS concluded that the findings were relevant for patients who meet COMS eligibility criteria [38] and who are suitable candidates for either enucleation or I-125 brachytherapy [38]. The COMS eligibility criteria regarding tumor size (2.5 mm to 10 mm apical height, no more than 16 mm in the longest basal diameter) were suitable in our case [38], however, our patient was using an immunosuppressive therapy (ibrutinib) for chronic lymphocytic leukemia (CLL) [15,16,38] and thus would have been excluded from the COMS study due to: (1) presenting a history of cancer; (2) using immunosuppressive therapy that could not be discontinued indefinitely [38].

Conservative surgery with local excision such as partial lamellar sclerouvectomy was not possible given the advanced mass size [39].

Enucleation was thus favorably evaluated with the hope for long-term prevention of metastatic disease, in consideration to the risk factors: extensive local iris invasion which repeatedly produced hyphema in a time span of 3 months (the first hyphema was in September, and the following episode was in December of the same year) and increased risk of hematogenous dissemination via the well-vascularized iris, tumor size (according to ultrasound measurement: the scan measured 11.84 mm), patient age (65 years old), difficulty in evaluating lymph node metastasis due to uveal melanoma in the presence of the patient’s CLL and inherent immunosuppression due to CLL and treatment with ibrutinib [15,16]. After thoroughly informing and consulting with the patient regarding prognosis and the available treatment options with possible outcomes, enucleation of the right globe was preferred as the best therapeutic decision and performed by the oncological surgeon.

The histopathology and prognosis of uveal melanoma is distinct from cutaneous melanoma. Broggi, G et al. [17] identified several prognostic factors. The subtype of the ciliary body uveal melanoma carries an additionally unfavorable prognosis compared to uveal melanomas confined to the choroid [14,17]. Inflammation signaled by lymphocytic infiltration (of >100 lymphocytes per 20 40× high power fields) [17] is often related to the loss of chromosome 3 and is a negative prognostic factor [17]. Microvascular features, evidenced using periodic acid Schiff (PAS) stain and classified by Foldberg et al. [17,40], are a dominant prognostic factor [40,41], with the study finding closed vascular loops in association with epithelioid cells and mitotic features as the most likely aspect to produce subsequent metastases [17,40]. A follow-up study using Factor VIII-related antigen (F8) reduced the microvascular patterns to five [41], and generally found a higher vessel count provided a poor prognosis, suggesting disordered growth and rapidly growing subclones [41]. Diffuse form [17], high mitotic index (mitoses/mm^2^) [17] and high Ki67 positivity are also unfavorable for overall prognosis [17]. Finally, tumor extension defined by invasion of adjacent sclera, greater thickness and large basal diameter greatly influence the overall prognosis [14,17]. Our case presented the subtype of ciliary body uveal melanoma which carries an additional unfavorable prognosis [17], and exhibited the presence of microvascular features with several blood vessels evidenced intratumorally (Figure 8 C,D), reported by Foss et al. [41] as a dominant prognostic factor. Inflammation signaled by lymphocytic infiltration was minimal below the threshold of >100 lymphocytes per 20 40× high power fields reported by Broggi, G et al. [17] as a negative prognostic factor. Ki67 was performed with resultant minimal positivity.

Risk stratification for the propensity to metastasize via genetic profiling for choroidal melanoma has been explored [17,42], with the results differentiating two classes: class 1 tumors are well-differentiated with low Ki-67 antibody positivity [17] (95% survival at 92 months [42]) and associate gain of 6p chromosome [42], while class 2 tumors are composed of ectodermal stem-cell-like cells and present high Ki-67 positivity [17] (31% survival at 92 months [42]) and associated loss of chromosome 3 [42], with upregulation of several genes [42].

Uveal melanoma can evade immune surveillance via multiple mechanisms such as the expression of inhibitory checkpoint programmed cell death ligand 1 (PD-L1), cluster of differentiation 47 (CD47), cluster of differentiation 200 (CD200) [43]. In another paper, Basile MS et al. [44] studied the effect of T-cell mediated inflammation on the expression of gene profiles involved in immune-escape tumor adaptation. Their study reported the increased expression of major histocompatibility complex (MHC) class I and II molecules, PD-L1 and programmed cell death ligand 2 (PD-L2) under inflammatory conditions [44], with interferon gamma (IFN-gamma) possibly responsible for upregulation of these molecules [44]. IFN-gamma has been reported to promote the survival of primary chronic lymphocytic leukemia (CLL) cells via JAK-Src/STAT3/Mcl-1 signaling pathway, with elevated IFN-gamma levels characteristic for advanced Rai stage disease [45].

Russo et al. [18] proposed that mutations of the cyclin-dependent kinase inhibitor 2A (CDKN2A) leading to its inactivation, through promoter methylation or loss of the 9p region, could play an important role in the development and metastatic progression of uveal melanoma, especially with 8q amplification. This mutation was found in one-third of uveal melanomas [18], and further studies are needed to better clarify the role of this gene in uveal melanoma progression, according to Russo et al. [18]. Mutations of the ubiquitin carboxyl-terminal hydrolase gene (BAP1) and loss of chromosome 3 are strictly related to uveal melanoma progression [18], with monosomy of chromosome 3 a predictor for ulterior metastases [18]. Lv Xiaohui et al. [46] noted that when BAP1 is mutated, it causes a significant risk of metastatic disease in uveal melanoma patients. Figueiredo CR et al. [47] linked the absence of BAP1 expression to an immunosuppressive tumor microenvironment. Lack of BAP1 increased the production of chemokines to attract T-cell aggregation, resulting in greater T-cell infiltration, a negative prognostic factor according to Broggi, G et al. [18]. On the other hand, mutations in eukaryotic translation initiation factor 1A X-Linked (EIF1AX) were found to be a protective factor in uveal melanoma metastasis [48].

In our case, the surgical treatment was successful, achieving curative effect without tumoral reoccurrence or evidence of metastatic disease for the following one and a half years. The patient could not attend the latest follow-up and further inquiry revealed that the patient was unable to obtain a protective level of COVID-19 antibodies after vaccination with a three-dose mRNA regimen, and thus was admitted to another medical service with severe COVID-19 disease. This unfortunate event exemplifies the profound immunosuppression associated with T cell disfunction caused by CLL. Clonal B cells in CLL uniquely interact with T cells in organized structures termed pseudofollicular proliferative centers (PC) [49] which are a hallmark of CLL and are not found in other B-cell neoplasms [49], resulting in weak stimulation leading to the generation and accumulation of CD4 central memory cells (TCM) [49]. Despite frequently elevated absolute numbers of circulating T cells [50], CLL clonal B-cells induce T cell anergy and improper Th2 polarization [4]. CLL patients present abnormal CD4 and CD8 T-cell phenotypes [50,51], with the increased frequency of T cells with the senescent phenotype [49], significant shortening of T cell telomeres [50] and an inversion of the CD4:CD8 ratio [52]. Furthermore, increases in specific T-cell subsets (CD8+ and CD4*PD-1 + HLA-DR+) versus healthy subjects is associated with CLL disease progression [52]. Noteworthy to our case, ibrutinib also modulates the immunosuppressive CLL microenvironment [16] and the innate immunity response in patients with COVID-19 infection [53]. This immunosuppression could pose higher risk to CLL patients for the development of immunogenic cancers such as melanoma [6,7], with a documented increased incidence of melanoma in CLL patients [6,7,8]. As such, we suggest that CLL patients benefit from regular screening for melanoma, which should also include ophthalmological screening via slit-lamp examination both without and with mydriatics. This would be effective in locating suspicious lesions located in the anterior or posterior pole, however, care must be taken regarding the posterior surface of the iris, which is not readily visible, and regarding ring melanoma which is difficult to diagnose with incipient symptoms mimicking pigment dispersion glaucoma. Valuable diagnostic clues are represented by the presence of sentinel vessels [30], localized iris rigidity exhibiting unequal dilation under mydriatic eye drops, and localized corneal oedema. If possible, or whenever warranted, such as in the presence of the abovementioned diagnostic clues, local ophthalmological imaging should be mandatory. UBM provides a high-resolution view of the angle and posterior iris surface [34], while mode B ultrasound is useful for posterior pole examination through opaque media such as cataracts. Providing an early diagnosis of uveal melanoma is established, local globe-preserving therapy can provide local disease control without necessity for enucleation. In selected cases, enucleation provides a curative effect for large masses or the development of metastatic disease, which has an unfavorable prognostic [11,32].

## 5. Conclusions

Our patient exhibited a “Masquerade syndrome” with ciliary body melanoma presenting as hyphema related to blood clotting abnormalities due to chronic lymphocytic leukemia disease. Upon further investigation, the bleeding was most probably caused by local invasion of the ciliary body melanoma into the vascularized iris, which both CLL and its treatment with BTK inhibitor ibrutinib could have accentuated. This paper thus underlines the importance of thoroughly screening patients with CLL for secondary malignancies which should also include detailed ophthalmological examinations due to the possibility of uveal melanoma. Furthermore, recurrent bleeding or even isolated bleeding in ophthalmological patients with CLL or undergoing treatment with BTK inhibitors or similar medication with mutagenic effect should always be suspect for ocular malignancy and warrant imagistic investigations such as UBM which provides a high-resolution view of the angle and posterior iris surface, or mode B ultrasound for posterior pole examination through opaque media such as cataracts. Such patients require extended follow-up care.

### 5.1. What Was Known

Patients immunosuppressed by CLL or undergoing treatment with BTK inhibitors present enhanced risk for the development of a secondary malignancy.

Uveal melanoma comprises a small percentage of melanoma, an immunogenic cancer, and can involve the iris, ciliary body or choroid, often involving multiple uveal structures.

### 5.2. What This Paper Adds

Ciliary body uveal melanoma can exhibit “masquerade syndrome” as hyphema in CLL patients.

Screening for secondary malignancy in CLL patients should include detailed ophthalmological screening with imaging using ocular mode B ultrasound or UBM, which offer visualization through opaque media or of the posterior plane of the iris.

Recurrent bleeding or even isolated bleeding in ophthalmological patients with CLL or undergoing treatment with BTK inhibitors or similar medication with mutagenic effect should always be suspect for ocular malignancy

## Figures and Tables

**Figure 1 diagnostics-12-01312-f001:**
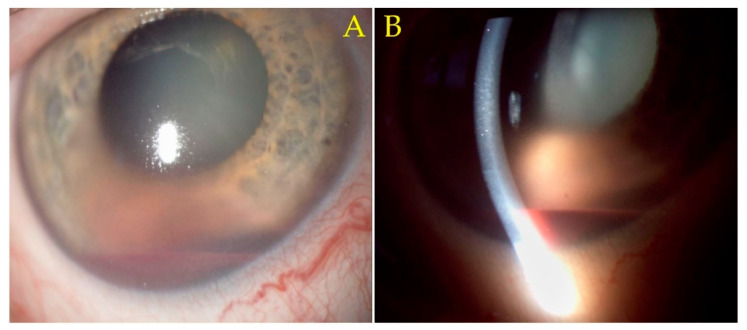
(**A**) The 12× slit-lamp biomicroscopy image. Irregular dark-pigmented mass, located inferiorly behind the pooling of blood (hyphema), involving the iris stroma and root, with adjacent corneal oedema. An enlarged conjunctival blood vessel is visible inferiorly, suggestive of sentinel episcleral vessel. (**B**) The 12× slit-lamp biomicroscopy image. A narrow slit-lamp highlights stromal corneal oedema and the hyphema.

**Figure 2 diagnostics-12-01312-f002:**
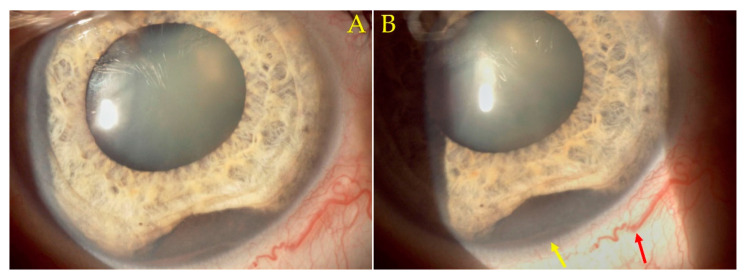
The 12× slit-lamp biomicroscopy images: (**A**,**B**) show absence of the normal iris stroma, with the presence of the pigmented tumoral mass at this level (**B**, **yellow arrow**). A likely sentinel blood vessel is present (**B**, **red arrow**).

**Figure 3 diagnostics-12-01312-f003:**
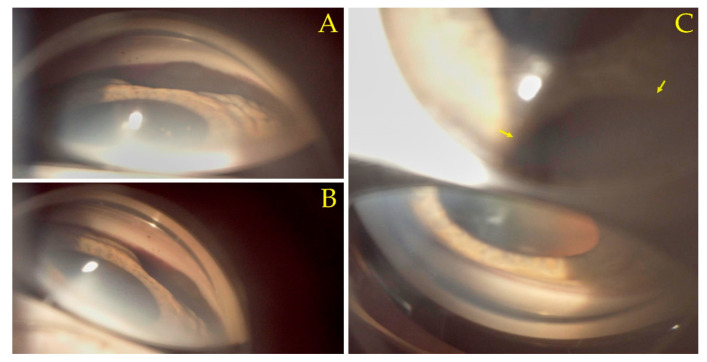
Three-mirror gonioscopy lens (Ocular Instruments, Bellevue, Washington State, United States of America) images visualizing the anterior chamber angle: (**A**,**B**) Presence of a tumoral mass at the inferonasal iris and anterior chamber angle level. (**C**) Shows via the lower mirror the normal superotemporal anterior chamber angle aspect (Shaffer grade 3 open angle). The tumor is visible through the central lens (**C**, **arrows**).

**Figure 4 diagnostics-12-01312-f004:**
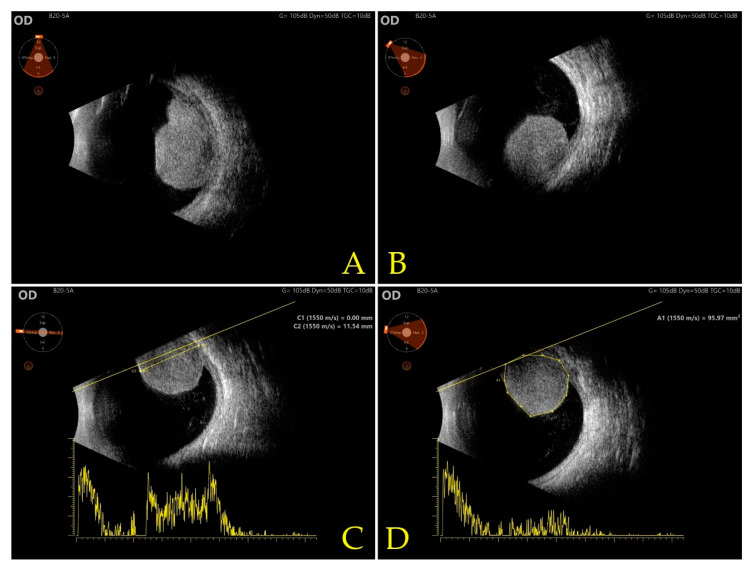
Mode B ultrasound (ABSolu Ultrasound, Quantel Medical, Cournon d’Auvergne, France) (**A**) inferior view, (**B**) inferior-nasal view, (**C**) measurement of anterior-posterior diameter 11.54 mm using T9-3 longitudinal view, (**D**) surface measurement 95.97 mm^2^ using nasal view.

**Figure 5 diagnostics-12-01312-f005:**
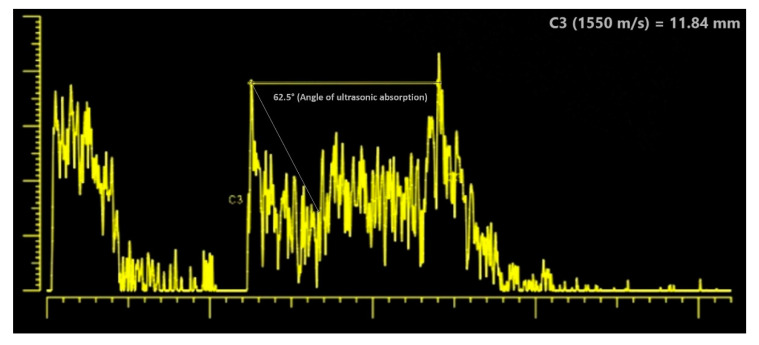
An ultrasound scan (ABSolu Ultrasound, Quantel Medical, Cournon d’Auvergne, France) with tumor measurement (C3 11.84 mm) and angle of ultrasonic absorption (62.5°).

**Figure 6 diagnostics-12-01312-f006:**
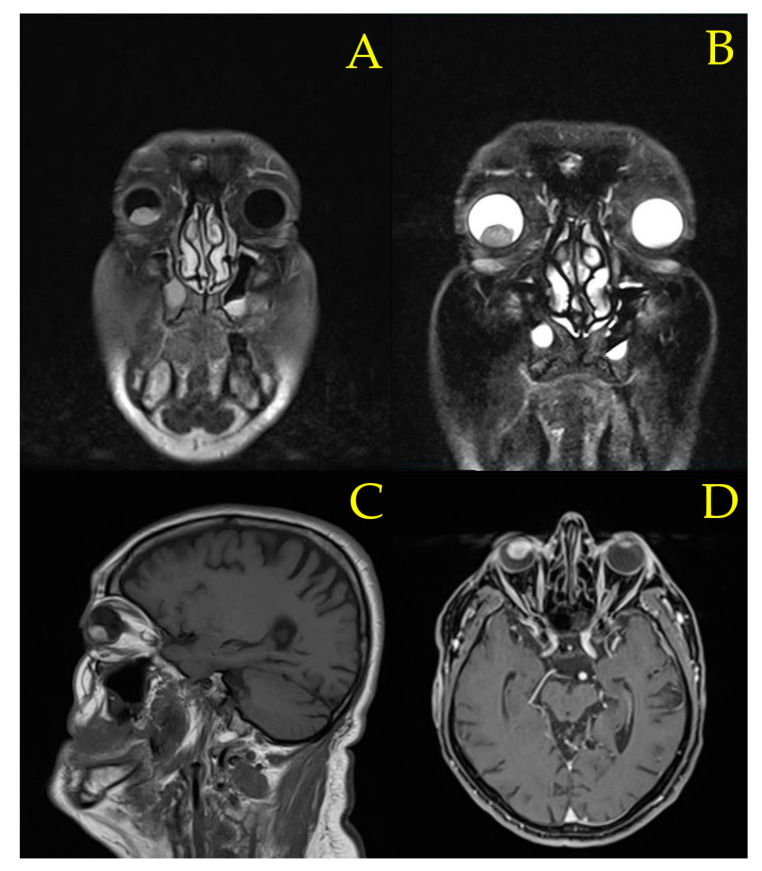
Contrast-enhanced magnetic resonance imaging (CE-MRI) evidencing an intraocular mass measuring 17/12 mm axial and 10 mm vertical with intense and homogenous contrast capture. (**A**) T2 turbo inversion recovery magnitude (TIRM) dark fluid. (**B**) T2 turbo spin echo (TSE) coronal thin slice with fat-suppression (FS). (**C**) T1 sagittal slice (**D**) T1 multiplanar reformation/reconstruction (MPR).

**Figure 7 diagnostics-12-01312-f007:**
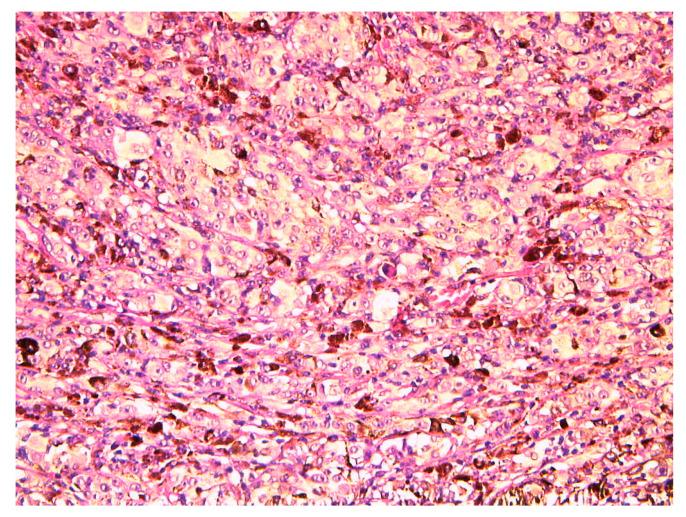
High magnification 20× hematoxylin eosin (HE) stain from the specimen revealing infiltration by epithelioid tumor cells and melanophages.

**Figure 8 diagnostics-12-01312-f008:**
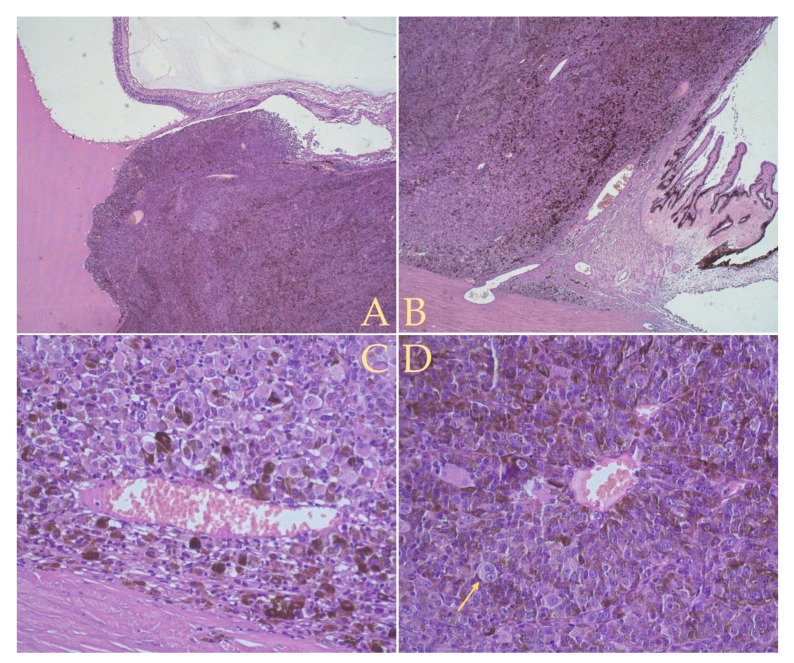
Hematoxylin and eosin (HE) stain: (**A**,**B**) were taken with a 5× lens and show overall tumor structure and adjacent invasion into the ciliary body. Melanic pigmentation is present relatively uniform. (**C**) The 40× lens image presents an enlarged tumor blood vessel, with erythrocytes in the lumen, surrounded by a melanocytic proliferation of intermediate morphology cells (in between spindle type B and epithelioid cells). (**D**) The 40× lens image presents tumor cells arranged into nests which are separated by fibrous conjunctival septa that converge towards a central blood vessel. Towards the bottom-left corner a large malign melanocytic cell presents two nuclei (**D**, **arrow**).

**Figure 9 diagnostics-12-01312-f009:**
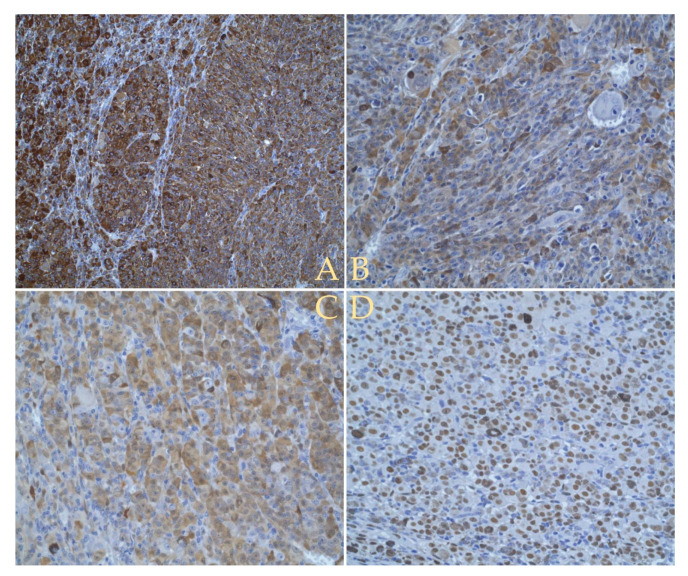
Histopathological exam using immunohistochemistry: (**A**) Image (20× lens) shows positive to staining using melanoma cocktail (HMB-45 + MART-1 − clones M2-7C10 + M2-9E3). (**B**,**C**) Images (40× lens) reveal staining with S-100 (EP32) antibody. (**D**) Image (40× lens) shows positive to Anti-SOX10 (SP267) antibody.

## Data Availability

All data regarding the patient’s clinical presentation, ophthalmological assessment and treatment undergone pertain to our ophthalmological department in the Clinical Hospital for Ophthalmological Emergencies, while data regarding the oncological assessment and surgical treatment of the patient also pertain to the Department of Oncology, Institute of Oncology Prof. Dr. Alexandru. Trestioreanu.

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
