# Peer review of "Underlying Ciliary Body Uveal Melanoma in a Patient with Chronic Lymphocytic Leukemia Presenting for Hyphema"

_diagnostics, 2022, doi:10.3390/diagnostics12061312_

Round 1

Reviewer 1 Report

Interesting topic but the research has not been conducted correctly. First of all the lesion has only been measured using Bscan ultrasound and CE-MRI but not using Ascan standardized ultrasound, a very important technique with diagnostic accuracy greater than 95% (reported by Ossoinig) and UBM. Furthermore, patient was not staged with TC total body.  It' wrong to say that I125 plaque brachytherapy it must be excluded as a therapeutic option due to the anterior localization of the lesion. It is an option with good local and systemic control. (Lumbroso -Leruic et al. I125 plaque brachytherapy for anterior uveal melanoma. Eye volume 18, pages911–916 (2004)).  Stereotactic radiosurgery offers an effective and safe approach for large uveal melanoma due to the satisfactory results in terms of local control, eye conservation and survival. According to COMS ( the collaborative ocular melanoma study) there are no difference in survival outcomes between patient underwent brachytherapy and enucleation. In this case a conservative option could be performed.

Author Response

Thank you for your review. Please see the attachment which contains the attached figure.

  1. A scan standardized ultrasound and UBM

Using mode A scan (attached figure 1) tumor diameter was measured at 11.84mm (C3) and angle of ultrasonic absorption was 62.5°. Ossoinig (Ossoinig KC, PMID: 395120) described four cardinal acoustic hallmarks of malignant melanoma on A-scan including a regular internal structure with similar height of the inner tumor spikes or regular decrease in height (positive angle kappa sign), low to medium reflectivity, solid consistency with no after movement of tumor spikes, and echographic sign of vascularization with a fast, spontaneous, continuous flickering vertical motion of single tumor spikes (Kaliki S, doi: 10.1038/eye.2016.275 PMID: 27911450) and according to Minning CA (PMID: 7114692), an ultrasonic angle of absorption between 40° and 65° degrees is a criteria for the ultrasound diagnosis of malignant melanomas. We could not perform a UBM examination at the moment of the patient’s admission to our service.

Attached figure 1 A Scan Ultrasound with tumor measurement (C3 11.84mm) and angle of ultrasonic absorption (62.5°).

  1. Staging with TC total body

Staging was reported as T2bNxM0 according to TC total body. The following enlarged lymph nodes were identified: right cardio-phrenic adenopathy, superior phrenic adenopathy with short axis under 8mm, lombo-aortic and several mediastinal (<6mm) enlarged lymph nodes. In our case, the patient’s chronic lymphocytic leukemia (CLL) comorbidity made assessing the presence of lymph node metastasis especially difficult due to enlarged lymph nodes caused by CLL. TC total body did not evidence hepatic metastasis.

  1. Stereotactic radiosurgery and I-125 Plaque brachytherapy as therapeutic options

Stereotactic radiosurgery and I-125 Brachytherapy are the other therapeutic options for uveal melanoma. Langmann G et. all (Langmann, G., Pendl, G., , Papaefthymiou, G., & Guss, H. (2000).   Gamma knife radiosurgery for uveal melanomas: an 8-year experience, Journal of Neurosurgery, 93(supplement_3), 184-188) noted that neovascular glaucoma can develop in patients after undergoing gamma knife radiosurgery for uveal melanomas located near the ciliary body and advised that such tumors be avoided and that the radiotherapy prescription dose be reduced to 40 Gy.  

According to COMS Randomized Trial of Iodine 125 Brachytherapy for Choroidal Melanoma (COMS Report No. 18, doi: 10.1001/archopht.119.7.969 PMID: 11448319) mortality rates among patients treated with eye conserving I-125 brachytherapy were similar to rates among enucleated patients for up to 12-years of follow-up and COMS concluded that the findings are relevant for patients who meet COMS eligibility criteria and who are suitable candidates for either enucleation or I-125 brachytherapy. The COMS eligibility criteria regarding tumor size (2.5mm to 10mm apical height, no more than 16mm in the longest basal diameter) are suitable in our case, however our patient was using an immunosuppressive therapy (ibrutinib) for chronic lymphocytic leukemia (CLL) and thus would have been excluded from the COMS study due to: 1) presenting a history of cancer; 2) using immunosuppressive therapy that could not be discontinued indefinitely (COMS Report No. 18, doi: 10.1001/archopht.119.7.969 PMID: 11448319).

In our case, the oncological surgeon identified the following risk factors for our patient: extensive local iris invasion which repeatedly produced hyphema in a time span of 3 months (the first hyphema was in September, and the following episode was in December of the same year) and increased risk of hematogenous dissemination via the well-vascularized iris, tumor size (according to ultrasound measurement: A scan measured 11.84mm), patient age (65 years-old) and difficulty in evaluating lymph node metastasis due to uveal melanoma in the presence of the patient’s CLL. Considering these factors, the oncological surgeon recommended enucleation as the treatment option, and after presenting all therapeutic options and associated risks to the patient, enucleation was preferred and performed.

Reviewer 2 Report

The authors presented an interesting case report of a 65-years-old patient affected by CLL and by an underlying, inferiorly situated, ciliary body uveal melanoma.

The manuscript is overall well written and the case well presented.

However, I have some concerns that must be addressed to improve the paper:

1. Histopathology of the present uveal melanoma case (was it a pure epithelioid cell melanoma? thickness? largest diameter? what was the pT stage of the tumor? Did you perform BAP-1 immunostaining on it?) and of uveal melanomas in general (prognostic factors, histopathologic and genetic features) must be better explained and discussed.

Please see:

PMID: 33154942

doi: 10.3390/app10228081

2. There are numerous genetic factors that can predict a good/poor outcome in terms of metastatic potential. Immune-escape genes have been studied as potential independent predictors of good prognosis. Please discuss this finding and see PMID: 30653520

Author Response

Thank you for your review. Please see the attachment which contains the attached figures.

  1. Histopathology of the present case

The enucleated right eye has a diameter of 25 mm. After sectioning the specimen the macroscopic aspect is of a pigmented tumor extending supero-anterior with invasion of the uveal tract. The largest tumor diameter is also the basal diameter of 12mm, and the apical height is 10mm. The distance between the tumor and the optical nerve is 12mm, without invasion of the optic nerve.

Histopathological microscopic analysis of the specimen (figure 1) revealed a predominant malign melanocytic proliferation of intermediate morphology cells (in between spindle type B and epithelioid, approximately 50%) with secondary proliferation of type B spindle cells (40%) and epithelioid cells (10%). According to American Joint Committee of Cancer (AJCC) classification (Kujala E doi: 10.1200/JCO.2012.45.2771 PMID: 23816968) our case presents a mixed cell type. The tumor cells are arranged into nests which are separated by fibrous conjunctival septa. Melanic pigmentation is present and relatively uniform intratumorally. A minimal chronic lymphocytic infiltrate is present, with melanophages situated mostly peritumorally. The proliferation of tumor cells invades the ciliary body, pars plana and choroid, with extension into the posterior chamber and secondary retinal detachment. Lympho-vascular or neural invasion was not detected. Maximal basal diameter was measured at 12mm, maximum height at 9mm.

Attached figure 1: Histopathological exam using hematoxylin and eosin (HE) stain: A and B are taken with a 5x lens and show overall tumor structure and adjacent invasion into the ciliary body. Melanic pigmentation is present relatively uniform. C (40x lens) presents an enlarged tumor blood vessel, with erythrocytes in the lumen, surrounded by a melanocytic proliferation of intermediate morphology cells (in between spindle type B and epithelioid cells). D (40x lens) presents tumor cells arranged into nests which are separated by fibrous conjunctival septa that converge towards a central blood vessel. Towards the bottom-left corner a large malign melanocytic cell presents two nuclei (D, arrow).

Staging was evaluated at pT2bNx.

Immunostaining was performed using melanoma cocktail HMB45 + MART-1 (clones M2-7C10 + M2-9E3) for melanoma screening, S-100 (EP32) antibody, Anti-SOX10 antibody (SP267), Ki67 Antigen (p3-NS1-Ag4-1) (figure 2). According to Russo et all (doi: 10.3389/fonc.2020.562074. PMID: 33154942) S100, SOX10 and HMB45 are significant markers of melanocytic differentiation and were positive in our case. After consulting with the histopathological department, we could not perform BAP-1 immunostaining due to insufficient sample size, however in response to your suggestion if it would be preferrable we could perform BAP-1 by PCR genetic determination by sending the sample to another laboratory, however after inquiry the laboratory requested at least 1 month for the results.

Attached figure 2: Histopathological exam using immunohistochemistry: A (20x lens) is positive to staining using melanoma cocktail (HMB-45 + MART-1). B and C (40x lens) reveal staining with S-100 antibody. D (40x lens) is positive to Anti-SOX10 antibody.

According to Broggi, G et all (PMID 33154942, doi.org/10.3390/app10228081) the subtype of ciliary body uveal melanoma, evidenced in our case, carries additional unfavorable prognosis compared to uveal melanomas confined to the choroid. Our case exhibited the presence of microvascular features with several blood vessels evidenced intratumorally (figure 1, C and D), reported by Foss et all (PMC1722145) as a dominant prognostic factor. Inflammation signaled by lymphocytic infiltration was minimal below the threshold of >100 lymphocytes per 20 40x high power fields reported by Broggi, G et all as a negative prognostic factor. Ki67 was performed with resultant minimal positivity.

  1. Genetic factors that can predict a good/poor outcome in terms of metastatic potential

Risk stratification for propensity to metastasize via genetic profiling for choroidal melanoma has been explored by Broggi et all (PMID 33154942, doi.org/10.3390/app10228081) and Onken et all (doi: 10.1158/0008-5472.CAN-04-1750. PMID: 15492234), with the results differentiating two classes: class 1 tumors are well-differentiated with low Ki-67 antibody positivity, while class 2 tumors are composed of ectodermal stem-cell-like cells and present high Ki-67 positivity. In their analysis Onken et all. reported only one metastatic death among class 1 patients, compared with eight in class 2 patients, and observed a 95% survival at 92-months for class 1 and 31% for class 2. Gain of chromosome 6p was observed in class 1 and correlated with a more favorable prognosis, while loss of chromosome 3 was characteristic for class 2 and negatively impacted prognosis. According to Broggi et all  analysis, Class 2 presented upregulation of E-cadherin (CDH1), Extracellular matrix protein 1 (ECM1), 5-Hydroxytryptamine (serotonin) receptor 2B (HTR2B), Rat Sarcoma Virus (RAS) related protein 31 (RAB-31) and downregulation of Eukaryotic translation initiation factor 1B (EIF1B), Fragile X syndrome autosomal homolog 1 (FXR1), Inhibitor of Deoxyribonucleic acid (DNA) binding 2 (ID2), LIM and cysteine-rich domains 1 (LMCD1), Leukotriene A4 hydrolase (LTA4H), Microtubule-associated tumor suppressor 1 (MTUS1), Roundabout, axon guidance receptor 1 (ROBO1), Special AT-rich binding protein (SATB) homeobox 1 (SATB1). Three control genes were reported: Mitochondrial ribosomal protein S21, Ribonucleic acid (RNA) binding motif protein 23 (RBM23) and Sin3A-associated protein, 130kDa (SAP130).

Uveal melanoma can evade immune surveillance via multiple mechanisms such as the expression of inhibitory checkpoints PD-L1, CD47, CD200 (Basile MS 10.3389/fonc.2019.01145. PMID: 31750244). In another paper Basile MS et all (doi: 10.1371/journal.pone.0210276. PMID: 30653520) studied the effect of T-cell mediated inflammation on the expression of gene profiles involved in immune-escape tumor adaptation. Their study reported increased expression of MHC class I and II molecules, PDL1 and PDL2 under inflammatory conditions, with IFN-gamma possibly responsible for upregulation of these molecules. IFN-gamma has been reported to promote survival of primary chronic lymphocytic leukemia (CLL) cells via JAK-Src/STAT3/Mcl-1 signaling pathway, with elevated IFN-gamma levels characteristic for advanced Rai stage disease. (Bauvois et all, doi: 10.3390/biomedicines9020188. PMID: 33668421).

Russo et all (doi: 10.3389/fonc.2020.562074. PMID: 33154942) proposed that mutations of CDKN2A leading to its inactivation, through promoter methylation or loss of the 9p region, could play an important role in the development and metastatic progression of uveal melanoma, especially with 8q amplification. This mutation was found in one-third of uveal melanomas, and further studies are needed to better clarify the role of this gene in uveal melanoma progression, according to Russo et all. Mutations of the BAP1 gene and loss of chromosome 3 are strictly related to uveal melanoma progression, with monosomy of chromosome 3 a predictor for ulterior metastases. Lv Xiaohui (doi: 10.3389/fimmu.2022.848455. PMID: 35309331) noted that when BAP1 is mutated, it causes a significant risk of metastatic disease in uveal melanoma patients. Figueiredo CR et all (doi: 10.1002/path.5384. PMID: 31960425) linked the absence of BAP1 expression to an immunosuppressive tumor microenvironment. Lack of BAP1 increased production of chemokines to attract T-cell aggregation, resulting in greater T-cell infiltration, a negative prognostic factor according to Broggi, G et all (doi.org/10.3390/app10228081, PMID 33154942,). On the other hand, mutations in EIF1AX were found to be a protective factor in uveal melanoma metastasis (Ewens KG doi: 10.1167/iovs.14-14550. PMID: 24970262).

Round 2

Reviewer 1 Report

Where do we understand the involvement of the iris? UBM was not performed. Was fluorescein iridography performed? 
Was there an iris bulging? (Figure 1 have a poor resolution).

In this study no genetic analysis was performed but only immunohistochemical and histological examinations therefore the discussion about the risk classes it must be reduced.

Author Response

Thank you for your review. In response to your requests, we attach the following.

  1. Involvement of the iris

Regarding the involvement of the iris and better-quality images, we took additional photos including gonioscopy lens images after the bleeding into the anterior chamber (hyphema) receded, at the 1-week follow-up. We attach images of the anterior pole (please see attached figure 1) showing absence of the iris stroma, having been substituted by the tumoral invasion. A likely sentinel episcleral blood vessel is present (attached figure 1, B red arrow).

Attached Figure 2 12x slit-lamp biomicroscopy images: A and B show absence of the normal iris stroma, with the presence of the pigmented tumoral mass at this level (B, yellow arrow). A likely sentinel blood vessel is present (B, red arrow).

We could not perform a UBM examination at the moment of the patient’s admission to our service or at the 1-week follow-up due to unavailability. Instead, after the bleeding receded, we performed a gonioscopy examination of the anterior chamber angle (please see attached figure 2), considered the clinical reference standard in assessing the anterior chamber angle (Cutolo, C.A. et. all,  https://doi.org/10.3390/diagnostics11122279). Gonioscopy (attached figure 2 A and B) revealed the presence of a tumoral mass at the infero-nasal iris and anterior chamber angle level. Iris bulging was not present, instead the tissue was replaced by the tumoral mass. For comparison C (attached figure 2) shows the normal angle structure in our patient viewed supero-temporal (grade 3 Shaffer open angle; Friedman DS, He M., doi: 10.1016/j.survophthal.2007.10.012. PMID: 18501270)

Attached Figure 3: Three mirror gonioscopy lens (Ocular Instruments, Bellevue, Washington State, United States of America) images visualizing the anterior chamber angle: A,B Presence of a tumoral mass at the infero-nasal iris and anterior chamber angle level. C shows via the lower mirror the normal supero-temporal anterior chamber angle aspect (Shaffer grade 3 open angle). The tumor is visible through the central lens (C arrows).

Regarding fluorescein iridography, in our country the contrast substance is considered by legislation a medical drug and is therefore subject to approval by the country’s national drug agency; unfortunately although the substance is available through suppliers, none have submitted to and completed the official approval process by the country national regulator body and as such, due to our work in a public, state-funded hospital, we could not use unapproved drugs such as contrast substance for performing fluorescein iridography.

  1. Reduction of the discussion about the risk classes

              We have reduced the paragraph with the discussion about the risk classes to the following:

Risk stratification for propensity to metastasize via genetic profiling for choroidal melanoma has been explored by Broggi et all (PMID 33154942, doi.org/10.3390/app10228081) and Onken et all (doi: 10.1158/0008-5472.CAN-04-1750. PMID: 15492234), with the results differentiating two classes: class 1 tumors are well-differentiated with low Ki-67 antibody positivity (95% survival at 92-months) and associate gain of 6p chromosome, while class 2 tumors are composed of ectodermal stem-cell-like cells and present high Ki-67 positivity (31% survival at 92-months) and associate loss of chromosome 3, with upregulation of several genes (PMID 33154942, doi.org/10.3390/app10228081).
